# The Role of Atropine in Preventing Myopia Progression: An Update

**DOI:** 10.3390/pharmaceutics14050900

**Published:** 2022-04-20

**Authors:** Alberto Chierigo, Lorenzo Ferro Desideri, Carlo Enrico Traverso, Aldo Vagge

**Affiliations:** 1IRCCS Ospedale Policlinico San Martino, University Eye Clinic of Genoa, 16132 Genoa, Italy; albertochierigo@gmail.com (A.C.); mc8620@mclink.it (C.E.T.); aldo.vagge@gmail.com (A.V.); 2Department of Neurosciences, Rehabilitation, Ophthalmology, Genetics, Maternal and Child Health (DiNOGMI), University of Genoa, 16126 Genova, Italy

**Keywords:** myopia, atropine, progression

## Abstract

Several approaches have been investigated for preventing myopia progression in children and teenagers. Among them, topical atropine has shown promising results and it is being adopted in clinical practice more and more frequently. However, the optimal formulation and treatment algorithm are still to be determined. We discuss the pharmacokinetic, pharmacodynamic, clinical, and tolerability profile revealed first by the multicenter, randomized ATOM 1 and 2 trials and, more recently, by the LAMP Study. Results from these trials confirmed the efficacy of low-concentration atropine with a concentration-dependent response. Although atropine at 0.025% and 0.05% concentrations has shown the most encouraging results in large-scale studies, these formulations are not yet commonplace in worldwide clinical practice. Moreover, their rebound effect and the possibility of reaching a stabilization effect have not been fully investigated with real-life studies. Thus, further larger-scale studies should better characterize the clinical efficacy of atropine over longer follow-up periods, in order to define the optimal dosage and treatment regimen.

## 1. Introduction

Myopia, also known as ‘nearsightedness’, is the one of the most common refractive diseases worldwide, and its prevalence is likely to rapidly increase in the near future [1,2]. Its onset usually occurs during childhood and it is caused by an excessive axial elongation of the eyes [3]. In some cases, myopia is a mere refractive error that can be corrected with spectacles, contact lenses, or refractive surgery. However, a smaller percentage of patients develop high myopia, which is currently defined by the World Health Organization (WHO) as a loss of six diopters (D) or greater. High myopia can lead to complications in the macula, in the peripheral retina, in the optic nerve, and in the lens and is therefore associated with an increased risk of blindness [4,5,6,7]. Moreover, myopic anisometropia may lead to amblyopia [8]. This shows the potential negative social and economic impact of myopia on healthcare systems all over the world.

Although we are still waiting for an evidence-based treatment algorithm for myopia, some strategies have shown a variable effectiveness in slowing its progression. These include more time spent outdoors [9,10], progressive addition lenses spectacles (PALs) [9,11,12], prismatic bifocal lenses spectacles (PBLs) [9,13], defocus spectacle lenses [14], soft contact lenses (SCLs) [9,15,16,17,18,19,20], orthokeratology (OK) [9,21,22,23,24,25,26,27,28,29,30], and various concentrations of antimuscarinic eye drops, mainly atropine, cyclopentolate, and pirenzepine [9]. These strategies are sometimes combined in order to increase their efficacy.

Time spent outdoors decreases the incidence of myopia in children, but its effect on the progression rate is insignificant [9,10]. PALs did not show a satisfactory efficacy [9,11,12]. PBLs led to a modest but significant decrease in axial elongation (AL) in myopic patients [9,13]. Of note, a significant reduction in myopia progression with minimal side effects was recently observed for defocus incorporated multiple segments (DIMS) spectacle lenses [14].

The results of OK have been encouraging in terms of clinical efficacy [21,22], but some concerns are represented by their possible drawbacks, including the risk of infectious keratitis and a relatively high dropout rate [23,24,25,26,27,28,29,30].

Finally, SCLs proved to have a modest efficacy in the slowing of myopia progression, albeit superior to spectacle lenses, especially for peripheral defocus modifying ones [9,15,16,17].

New developments in this field include MiSight contact lenses, whose efficacy was demonstrated in at least two clinical trials [18,31].

To date, atropine has shown promising results in preventing myopia progression. In particular, lower doses of atropine have revealed the most advantageous in balancing between clinical efficacy and the low rate of adverse effects, even though a rebound phenomenon after the interruption of the treatment has been reported [9]. Pirenzepine (an M1-selective antimuscarinic) and oral 7-methylxantine (an adenosine antagonist) have also been reported to slow myopia progression in children [32,33].

In this review, we discuss the role of topical atropine for preventing myopia progression in children and adolescents. We also describe its pharmacokinetics, pharmacodynamics, safety, tolerability, and clinical efficacy.

## 2. Materials and Methods

A literature search was conducted to find all the published studies regarding the pharmacokinetics, pharmacodynamics, clinical efficacy, and safety of atropine for treating myopia, from inception until 2021. The following electronic databases were used: Medline, PubMed, Science Citation Index via Web of Science, and the Cochrane Library. The following search terms were used: “atropine”, alone or in combination with ‘myopia’, ‘adolescents’, ‘children’, ‘clinical efficacy’, ‘safety’, ‘toxicity’. Current clinical trials on the efficacy and tolerability of atropine were explored on research registers such as http://www.clinicaltrials.gov/ (accessed on 21 March 2022). Moreover, all the articles and their reference lists were thoroughly analyzed in order to find other manuscripts that could be included in this drug evaluation.

## 3. Results

### 3.1. Antimuscarinic Eye Drops: Overview of the Market

Antimuscarinic drops are used all over the world as cycloplegics, mydriatics, and for the penalization of the healthy eye in the treatment of amblyopia [34]. In the last decades, we have witnessed an increase of evidence proving the efficacy of antimuscarinic drugs in preventing myopia progression in children all over the world. However, their ocular topical use is still off-label in many countries. In fact, no pharmaceutical agent has been approved by the US FDA for preventing myopia progression, although atropine is already used in Asia to control myopia in children [35]. The effectiveness in preventing myopia progression in children has been demonstrated for different concentrations of atropine, cyclopentolate 1%, and for pirenzepine 2% [9]. Several randomized trials and meta-analyses have explored both the efficacy and the side effects of different atropine concentrations. The impact on AL and SED was highest for 1% atropine and lowest for 0.1% [35,36,37]. However, higher concentrations were also less tolerable and had a higher incidence of rebound effect after treatment discontinuation [34].

Currently, clinicaltrials.gov lists 19 clinical trials that are testing the effectiveness of various concentrations of atropine in slowing myopia progression, among which 11 are currently recruiting.

We list the following ones:The Use of Atropine 0.01% in the Prevention and Control of Myopia (ATOM3), in which children with family history of myopia will be randomized to atropine or placebo. They will be treated for 2–2.5 years, followed by one year of washout [38].Microdosed Atropine 0.1% and 0.01% Ophthalmic Solutions for Reduction of Pediatric Myopia Progression, which will last 48 months. Children will be randomized to receive either atropine 0.1%, 0.01%, or placebo, evaluated at regular intervals for 36 months and then re-randomized and followed for an additional year [39].Topical 0.01% Atropine for the Control of Fast Progressing Myopia (Myopie-STOP), with the aim of evaluating the efficacy at 1 year of 0.01% atropine on the reduction of fast progressing myopia in children aged 4–12, compared to a placebo [40].

Currently, no ongoing trials investigating the role of cyclopentolate or pirenzepine have been found.

### 3.2. Atropine: Introduction to the Compound

Chemical formula

The chemical formula of atropine is the following: C_17_H_23_NO_3_.

### 3.3. Pharmacokinetics

The literature about the pharmacokinetics of atropine eyedrops is scarce. Topical atropine has a partition coefficient of 1.83 and a pKa of 9.43 at 7.4 pH, which means that it is ionized on the ocular surface [41,42]. A recent pharmacokinetic study on rabbits revealed that after 5 h from topical administration, the highest concentration of atropine was detected in the conjunctiva, with a concentration gradient established anteriorly to posteriorly. Moreover, the concentrations in the cornea and sclera were similar. Therefore, the authors concluded that atropine reaches the anterior and posterior chambers by simple diffusion via the conjunctival, scleral and uveal routes [43]. After 24 h, preferential binding of atropine to posterior ocular tissues was found. Atropine showed a good ocular bioavailability with concentrations of two magnitudes higher than its binding affinity in most tissues after 3 days [43]. It has been also reported that atropine binds melanin, both in vitro and in rabbits [44].

In humans, the systemic absorption of topically applied atropine is generally low, but systemic side effects can indeed occur, especially in children, likely due to their smaller body volumes. In one study the reported systemic bioavailability of atropine in healthy individuals ranged from 19% to 95% [45]. The largest amount of the drug is metabolized by enzymatic hydrolysis, particularly in the liver and 13–50% of the molecule is excreted unmodified in the urine [46,47].

Pharmacokinetic studies highlighted that the pharmacological effect begins after 48–120 min from its administration and lasts until 7–14 days [48].

### 3.4. Pharmacodynamics

Atropine is a nonselective reversible muscarinic antagonist. It binds to all five subtypes of muscarinic receptors (mAchR, MR1–MR5), preventing acetylcholine from interacting with them. On the other hand, pirenzepine is M1-selective [49,50]. These receptors are coupled with G-proteins (GPCR) and have been found in the human iris, ciliary body [51], lens epithelium [52], retinal amacrine [53] and pigment epithelial cells [54,55], and scleral fibroblasts [56].

The pathological mechanism of myopia and the pathways involved in the antimyopic effects of atropine are still largely unknown. However, some evidence on the matter does exist.

For example, Lind et al. studied chick fibroblasts in vitro and reported that the antimyopic effect of antimuscarinics may be at least partially mediated by the inhibition of M1 receptors in the sclera [57]. A study in chicks demonstrated the inefficacy of M2 antagonists in opposing form deprivation myopia [58]. However, M2 receptors have been implicated in myopia development in an in vivo study on mice [59]. It was also reported that a M4-antagonist can prevent myopia progression in chicks [60]. Therefore, one of the pathways for the antimyopic effect of antimuscarinics is the interaction with ocular mAchR.

A study in chicks reported that neither cholinergic amacrine cells nor mAchR were necessary for the induction of form-deprivation myopia and for the antimyopic activity of atropine, which means that other cellular pathways may be involved [61].

In this regard, a protective role of dopamine (DA) has long been described [62], but a comprehensive theory for its mechanism of action is still lacking [63]. In the human retina, dopamine is produced by amacrine and interplexiform cells [64]. Dopamine has five GPCRs (D1–D5), some of which have been identified in animal retinal cells and RPE [65,66,67]. In mice, the activation of D1 inhibited the development of myopia [68,69]. In tree shrews, the activation of D2 and D4 receptors was reported to have an antimyopic effect [70], but in guinea pigs and D2 knock-out mice the activation of the D2 receptor seemed to do the opposite [71,72]. Moreover, nonselective D-agonists can inhibit myopia progression in animal models [73,74]. This leads us to the conclusion that atropine and DA could act in parallel biochemical pathways that would later converge on a common effector [75].

In particular, in preclinical models, atropine has been shown to stimulate the release of DA into the extracellular space and the vitreous, which may inhibit a retinal signaling process that is supposed to be involved in axial elongation, and thus myopia progression [76].

Furthermore, it has been shown that dopamine could act directly on the cornea, as some dopaminergic receptor activity is located in rabbit and bovine corneas.

Moreover, other cellular pathways for the antimyopic effect of atropine have emerged, such as the inhibition of human alpha2A-adrenergic receptors (in vitro) [77] and nitric oxide signaling [78,79].

Since the accommodative hypothesis has long been rejected [80], the posterior segment currently seems to be where atropine exerts its antimyopic effects.

Some preclinical studies have reported that atropine induces extracellular matrix (ECM) biosynthesis in scleral fibroblast cells, therefore thickening the scleral tissue and decreasing its elasticity and tendency to elongation. Furthermore, an in vitro study on human cells proved that atropine was able to decrease in vitro ECM biosynthesis in the choroid, with local fibroblasts improving scleral blood perfusion through the choroid, improving ECM perfusion and thus decreasing myopia progression [81]. Furthermore, atropine has been able to increase choroidal thickness in both healthy and myopic children and to block choroidal thinning associated with hyperopic defocus [82,83,84,85]. Future research could provide us with an exhaustive theoretical model that would explain in detail the biological pathways and effectors that determine the observed efficacy of this drug.

### 3.5. Clinical Efficacy

#### 3.5.1. The ATOM 1 and ATOM 2 Studies

In the last few decades, low concentrations of atropine have gained considerable attention for their efficacy in slowing myopia progression in children. This was initially reported in non-randomized studies [86,87,88].

Subsequently, the Atropine for the Treatment of Childhood Myopia study (ATOM 1), which was a two-year long, randomized, placebo-controlled, double-masked trial, demonstrated that the progression of myopia was substantially slower in the group treated with atropine 1% than in the one that received a placebo. Specifically, 65.7% of atropine-treated eyes had a progression of less than 0.50 D, and 13.9% of them progressed more than 1.00 D. In contrast, 16.1% and 63.9% of placebo-treated eyes showed a progression of less than 0.50 D and more than 1.00 D, respectively [35]. Moreover, after two years, the treated group had a significantly slower myopia progression, with a difference of −0.92 D in SED (95% confidence interval (CI): −1.10 to −0.77 D; *p* < 0.001) and 0.40 mm in AL (95% CI: 0.35–0.45 mm; *p* < 0.001) compared with the placebo.

In 2009, Tong et al. re-evaluated the same patients of ATOM 1 for myopia progression after a one-year washout period. The authors reported an overall reduction in myopia progression since the beginning of ATOM1, but observed a higher rate of progression during the washout period in the treatment group compared with the placebo one, especially in the first six months [89].

Specifically, after the whole three years (two of treatment, one of wash-out), the SED was −4.29 ± 1.67 D (*p* < 0.001) in the atropine 1% group vs. −5.22 ± 1.38 D (*p* > 0.001) in the placebo group. During the wash-out year, the rate of myopia progression in the atropine-treated group was −1.14 ± 0.80 D (*p* < 0.001), while in the placebo group it was −0.38 ± 0.39 D (*p* < 0.0001) [90].

Subsequently, the Atropine for the Treatment of Childhood Myopia 2 (ATOM 2) study was published, which had the aim to assess whether lower concentrations of atropine could be effective in reducing myopia progression, with potentially fewer side effects. The study comprised a two year-long treatment phase, followed by a one-year-long washout period.

Patients were randomized into three treatment groups (atropine 0.5%, atropine 0.1% and atropine 0.01%). After two years, myopia progression was lowest in the group treated with atropine 0.5% and highest in the 0.01% one. However, the progression rate of patients treated with atropine 0.01% was deemed clinically not very different from that seen at higher concentrations, and the ocular side effect profile was significantly better. Indeed, 50% of the 0.01% group progressed by less than 0.5 D, while these rates were 58% and 63% in the 0.1% and 0.5% groups, respectively. Among all three groups, around 18% of subjects progressed by 1.0 D or more [91].

In the atropine 0.5%, 0.1%, and 0.01% groups, the average myopia progression were −0.30 ± 0.60 D, −0.38 ± 0.60 D, and −0.49 ± 0.63 D, respectively (*p* = 0.02 for the 0.01% and 0.5% groups; *p* > 0.05 for the other concentrations), and the AL increased by 0.27 ± 0.25, 0.28 ± 0.27 and 0.41 ± 0.32, respectively (*p* < 0.05).

The ATOM 2 patients were then studied after a one-year-long washout period (ATOM 2, phase 2), confirming a dose-related rebound phenomenon. The progression was fastest (−0.87 ± 0.52 D, *p* < 0.001) in children treated with atropine 0.5% and slowest in the 0.01% group (−0.28 ± 0.33 D, *p* < 0.001). The 0.1% group had a progression of −0.68 ± 0.45 D (*p* < 0.001). AL growth was 0.35 ± 0.20 mm in the 0.5% group, 0.33 ± 0.18 mm in the 0.1% group, and 0.19 ± 0.13 mm in the 0.01% group (*p* < 0.001). Overall, this negated the initial superior effects of 0.5% and 0.1% atropine compared to 0.01%. Indeed, in the whole 36 months, the progression was −0.72 ± 0.72 D, −1.04 ± 0.83 D, and −1.15 ± 0.81 D in the 0.01%, 0.1%, and 0.5% groups, respectively [36,90].

After these results, all ATOM 2 patients who had a myopia progression rate higher than −0.50 D during the washout phase received a treatment with atropine 0.01% for an additional two-year period (ATOM 2, phase 3), which successfully slowed the progression of myopia. The progression over the whole five years was −1.38 ± 0.98 D, −1.83 ± 1.16 D, and −1.98 ± 1.1 D (*p* < 0.001) in the 0.01%, 0.1%, and 0.5% groups, respectively [91]. This further demonstrated that atropine 0.01% drops had the best overall clinical profile, since they caused the least rebound phenomenon, had the best long-term clinical efficacy, and the most tolerable side effects of all the investigated doses.

A third version of this trial, ATOM 3, began in 2017 and is expected to end in 2023 [38].

#### 3.5.2. Other Relevant Clinical Studies

The advantage of lower atropine concentrations was investigated by Shih et al. in 1999, who randomized 186 children for two years in four treatment groups: three received different concentrations of atropine (0.5%, 0.25%, 0.1%) and the fourth 0.5% tropicamide, which served as a control [92]. The mean annual myopic progression was 0.04 ± 0.63 D, 0.45 ± 0.55 D, 0.47 ± 0.91 D in the 0.5%, 0.25%, 0.1% atropine groups, respectively (*p* < 0.01). In the tropicamide group, it was 1.06 ± 0.61 D (*p* < 0.01) [92].

One of the most significant limitations of ATOM 1 and ATOM 2 is their possible selection bias, since only children of Asian ethnicity were investigated. For this reason, Polling et al. studied the efficacy of atropine 0.5% in a sample of Dutch children with high myopia, all of whom were of European, Asian, or African descent. The study was not randomized, but nonetheless confirmed the efficacy of this treatment, with a decrease in myopia progression rate from −1.0 D/year ± 0.7 before the treatment to −0.1 D/year ± 0.7 after 12 months [93].

Subsequently, Polling at al. recruited a total of 124 patients and conducted a prospective clinical efficacy study for three years using atropine 0.5%. This time the mean annual spherical progression and increase in AL were −0.25 D (inter quartile range (IQR): 0.44) and 0.11 mm (IQR: 0.18), respectively. In good responders the atropine concentration was gradually lowered every six months, provided the myopia progression was stable. The atropine concentration was increased if the progression was moderate to insufficient [94].

Clark et al. extended these results to American children, treating them with atropine 0.01% for one year in a retrospective, case-control study. This time the decrease in myopia progression was −0.1 ± 0.6 D in the treated group, vs. −0.6 ± 0.4 D in controls [95].

In a cross-sectional, single center, observational case series, Joachimsen et al. reported a decrease in myopia progression in children after one year of atropine 0.01% use. The mean progression before the treatment was −1.05  ±  0.37 D per year, which fell to −0.40  ±  0.49 D after the treatment (*p* <  0.0001) [96].

Wu et al. conducted a retrospective, case–control study with a minimum follow-up of three years, from 1999 to 2007, where children received 0.05% atropine if they had a progression over −0.5 D after 6 months. The concentration was raised to 0.1% if the progression was still higher than −0.5 D after another six months. Untreated children served as controls. The adjusted myopia progression in the treatment and control groups were −0.23 D/year and −0.86 D/year, respectively (*p* < 0.001) [97].

In 2020 Zhu et al. tried to assess the effect of 1% atropine vs. placebo in an effectiveness study, which had a prospective, clinic-based and placebo-controlled design. The subjects received either atropine 1% or saline once a month for two years, then every two months for one year, then no treatment for one year. At the end of the four years, the corrections needed for the treatment and placebo groups were −4.96 ± 1.22 D and −7.28 ± 1.26 D, respectively (*p* < 0.001). The AL were 25.48 ± 0.29 mm and 26.59 ± 0.20 mm, respectively (*p* < 0.01). In addition, the mean progression per year in the treatment group was lower than that in the control group, respectively, −0.29 ± 0.17 D vs. −0.89 ± 0.44 D, (*p* < 0.05) [98]. Diaz-Llopiz et al. conducted a five-year-long randomized, placebo-controlled study, showing a slower myopia progression in 200 children aged 9–12 years treated with atropine 0.01% daily. The progression rates in the treatment and placebo groups were −0.14 ± 0.35 and −0.65 ± 0.54, respectively (*p* < 0.01) [99].

#### 3.5.3. The LAMP Study

The Low-concentration Atropine for Myopia Progression (LAMP) Study consisted of a double-blinded, placebo-controlled clinical trial which was conducted in order to determine the best atropine concentration among 0.05%, 0.025%, and 0.01%. A fourth group of children received a placebo [37]. During the first year (phase 1) 0.05% atropine showed the best treatment-side effect ratio. Since the ATOM 2 study reported an increase in the efficacy of 0.01% atropine during the second year, patients were studied for an additional year of treatment (phase 2), and those in the placebo group began a treatment with atropine 0.05% (switchover group). A concentration-dependent response was observed, and 0.05% atropine continued to be the most effective concentration in the whole two years of the study. Indeed, after the second year, the mean SE change was −0.55 ± 0.86 D, −0.85 ± 0.73 D, and −1.12 ± 0.85 D, respectively for the 0.05%, 0.025%, and 0.01% groups (*p* = 0.015, *p* < 0.001, and *p* = 0.02). The AL changes were 0.39 ± 0.35 mm (*p* = 0.04), 0.50 ± 0.33 mm (*p* < 0.001), and 0.59 ± 0.38 mm (*p* = 0.10), respectively in the 0.05%, 0.025%, and 0.01% groups.

Moreover, the authors reported that the efficacy of 0.05% atropine in the LAMP study was similar to that of 0.01% atropine in the ATOM 2 trial, and that 0.01% atropine showed a lower effect in the LAMP study than in ATOM 2 study. At two years, 52.7%, 32.0%, and 22.0% of subjects in the 0.05%, 0.025% and 0.01% atropine group progressed less than 0.5 D, respectively, and 9.1%, 7.0%, and 19.2% had a SE progression of 2D or more, respectively.

The third phase of the LAMP study is still ongoing. The patients in each arm will be randomized into either a wash-out group or a continued treatment group. This would allow the authors to investigate the efficacy of each atropine concentration over three years, as well as the possible rebound phenomenon [100].

The fourth phase is designed to investigate the long-term efficacy of low concentration atropine after five years. It will consist in the resumption of atropine in children who progressed during the washout period [100] (Table 1 and Table 2).

### 3.6. Safety and Tolerability

Topical atropine is associated with some ocular AEs, including mydriasis, photophobia, and reduced accommodation, with symptoms of glare and blur during near-work activities. Although usually mild, side effects can hinder school and outdoor activities, and therefore represent a significant cause of treatment interruption.

The most problematic AEs reported with atropine eye drops are the rebound phenomenon and ocular side effects, which occur in around 5% of patients. The former is defined as the rapid increase in axial elongation shortly after treatment discontinuation. Other relevant side effects are an ocular allergic reaction and photophobia and blurriness of near vision, which are due to myosis and cycloplegia [35,37,94,98,99]. The ocular side effects are more common and marked at higher concentrations of the molecule. This is especially true for the rebound phenomenon, which seems to be dose-related. In addition to this, low concentrations of atropine are often available in formulations that use benzalkonium chloride (BAK) as a preservative, which is known to be toxic to the corneal epithelium and it has been associated with dry eye syndrome. Some studies have even highlighted its toxic effects to the retinal tissue [101,102].

We now present more detailed list of the side effects in each of the aforementioned studies.

The ATOM 1 study reported only ocular side effects, which were allergic reactions (4.5%), glare (1.5%) and blurred near vision (1%) [35]. During the ATOM 2 study, the rebound phenomenon appeared to be dose-related [36].

Polling et al. observed a high incidence of side effects, including photophobia, reading problems, and headaches [93].

Although they could measure a small difference in pupil size, Joachimsen et al. did not report any clinically significant side effect [96]. Zhu et al. did not report any serious AE, even though 62.12% of children experienced photophobia, 19.70% blurred near vision, 18.5% eye irritation, 5.451% conjunctivitis, blepharitis, and 0.9% a local allergic reaction [49,98]. In the study conducted by Diaz-Llopez et al., 2% of children reported photophobia, reading problems and headaches, which led to treatment interruption [99].

Polling et al. found that a small percentage of subjects had side effects that led to treatment discontinuation. To be more specific, 6.8% had an allergic reaction and 13.6% photophobia and non-eye-related AEs [94].

In the LAMP study, the main AEs were photophobia, blurred near vision, and allergic conjunctivitis. The incidence of the latter was similar at all atropine concentrations and was never the cause of hospitalization. At two years, photophobia was experienced by 8.6%, 4.7%, and 5.5% of patients in the 0.05%, 0.025%, and 0.01% groups, respectively. Distance and near vision were similar in all groups, as was the vision-related quality of life [37].

Systemic AEs are uncommon and include dry mouth, face flush, headache, high blood pressure, constipation, difficulty in urination, and central nervous system disturbances. Although they have not yet been reported in the aforementioned studies, should they occur, they can be treated with intravenous physostigmine [48].

### 3.7. Regulatory Affairs

Even though various concentrations of atropine have long been FDA-approved and available on the market, their use for slowing myopia progression in children and adolescents is still off-label and experimental. However, the studies published in the last few decades and presented here do show promise, and we could therefore expect to witness a change in this regard as a worldwide standardized approach emerges.

## 4. Discussion

Several therapeutic approaches have been developed to slow down the progression of myopia in children and adolescents. Atropine drops appear to be among the most effective [9], even if the exact mechanism of action remains to be clarified and it is not free of drawbacks, including the rebound effect and other ocular AEs.

A network meta-analysis published by Huang et al. compared the efficacy of a number of treatment strategies in slowing myopia progression in the youth, which included high (0.5% and 0.1%), moderate (0.05%), and low (0.01%) concentrations of atropine [9]. Among all of them, atropine was the most effective, and this was particularly true for high (1% and 0.5%) concentrations of this drug, which were significantly superior to other interventions, except for moderate-dose atropine (0.1%) and low-dose atropine (0.01%) [9]. This meta-analysis suggested that atropine 0.01% could be considered the best treatment option for preventing myopia progression in children and adolescents, because of its minimal side effects and least rebound phenomenon.

Afterwards, the LAMP Study suggested that atropine at 0.05% was the most effective formulation, slowing the progression by 0.54 D and AL by 0.21 mm after 1 year. By contrast, atropine at 0.01% resulted clinically ineffective by slowing the progression only by 0.22 D and AL by 0.05 mm. Hence, results from the LAMP Study confirmed the efficacy of low-concentration atropine with a concentration-dependent response in comparison with placebo. In this regard, an important limitation of the study is represented by the lack of a placebo group after the second year of treatment with the administration of atropine 0.05% (due to ethical reasons) [37]. Another remaining issue is the rebound phenomenon observed after the cessation of atropine 1%, 0.5%, 0.1%, and 0.01%, as observed in the ATOM 1 and ATOM 2 studies [35,36]. In fact, it has been postulated that the continuous administration of atropine for two years may be associated with a stabilization effect, with the possibility of stopping the therapy afterwards [37]. Further clinical trials are needed in order to investigate this issue, considering the important clinical implications in terms of tailored treatment algorithms.

Furthermore, despite the evidence provided by the LAMP Study that 0.01% atropine is not clinically effective in slowing AL, this formulation is being largely administered in real-life clinical practice and this represents potentially alarming data [103]. For this reason, considering the overall better clinical results of atropine 0.025% and 0.05%, further clinical trials should provide more evidence about this formulation, in order to develop an effective and widely accepted treatment algorithm, since some treatment strategies have already emerged [104].

Of note, the long-term side effects of atropine eye drops have not yet been evaluated [105]. The World Health Organization currently recommends limiting treatment to two years [4].

Effective alternatives to atropine are starting to emerge, too. A two year-long, double-masked, randomized clinical trial reported that DIMS spectacle lenses significantly reduced myopia progression with minimal side effects [14]. Specifically, the overall spherical progression was −0.41 ± 0.06 D in the DIMS group and −0.85 ± 0.08 D in the single vision spectacles (SV) group. The total increase in AL was 0.21 ± 0.02 mm and 0.53 ± 0.03 mm, respectively for the DIMS and SV groups. However, the authors reported an inferior efficacy compared with atropine eyedrops [14].

Another recent alternative to atropine are MiSight contact lenses, whose efficacy was demonstrated in at least two clinical trials [18,31]. Chamberlain et al. studied them in a three-year-long, randomized, double-masked and placebo-controlled study [18]. After three years, the changes in cycloplegic spherical equivalent refraction in the MiSight group were 0.73 D D less than in the control group (a 59% difference), and the axial elongation was 0.32 mm less (a 52% difference) [18]. The lenses were well-tolerated and had few side effects. Moreover, 41% of patients in the MiSight group had no progression in spherical equivalent refraction. In another trial, no rebound effect was observed for MiSight lenses after one year [19].

Moreover, certain nutritional measures have been beneficial for some eye diseases, and a high BMI was associated to high myopia in the KNHANES VII study [106,107,108]. It could be interesting to gain more insight on the role of diet and supplements in preventing or slowing myopia progression.

The main advantages of atropine rely on its relatively safe and effective profile. A comparison between the potential cost of these interventions has not been published yet.

Several treatment algorithms have been proposed in myopic children in order to slow the diseases. However, to date, none of them has gained worldwide acceptance. A reasonable approach could be to initially treat progressing children with an evidence-based pharmacological or optical approach. If progression is halted, the optical treatment could be interrupted or the drug titered-down. Combined treatments and higher drug concentrations could help decrease progression in non-responders and fast-progressors.

This way, the incidence of high myopia could decrease, and with it the frequency of myopia-related complications.

In the next years, a more detailed treatment algorithm with the adoption with the optimal dosage in relation to the risks/benefits ratio should be developed and for this reason further, larger scale trials with longer follow-up periods are needed.

## 5. Conclusions

Atropine has been long known for its properties as a mydriatic and cycloplegic. Recently, a new side to this molecule has emerged, which is its potential in preventing myopia progression in children and adolescents [35,36,37]. However, the exact mechanism of action of this drug and the pathophysiology of myopia onset and progression remain to be clarified.

In any case, atropine has shown promise in preventing the progression of myopia in children, and we are constantly transitioning to a concentration that has the highest efficacy and fewest side effects.

Indeed, despite being more effective, higher concentrations (1%, 0.5%) are associated with an increased risk of a rebound phenomenon and ocular side effects [35,36]. On the other hand, lower concentrations (in particular 0.05%) seem to maintain a clinically satisfying efficacy, with a much lower incidence of ocular side effects [9,37]. As a consequence, low concentrations of atropine appear to have a better clinical profile and represent a valid treatment strategy to slow the progression of myopia in children and adolescents. However, further randomized clinical trials with larger follow-up periods are needed.

## Figures and Tables

**Table 1 pharmaceutics-14-00900-t001:** Clinical trials on the efficacy of atropine in controlling myopia progression.

Study Name	Design	Demographic		Follow-Up	Main Outcomes	
		Number of Patients	Mean Age (and Interval, in Years)		SEP (D)	AL (mm)
ATOM 1 [90]	Randomized, placebo- controlled, double-masked	400	9 (6–12)	2 years	−1.20 ± 0.69 (placebo) −0.28 ± 0.92 (A: 1%)	0.38 ± 0.38 (placebo) −0.02 ± 0.35 (A: 1%)
ATOM 2 [91]	Randomized, double-masked	400	10 (6–12)	2 years	−0.49 ± 0.60 (A: 0.01%) −0.38 ± 0.60 (A: 0.1%) −0.30 ± 0.63 (A: 0.5%)	0.41 ± 0.32 (A: 0.01%) 0.28 ± 0.27 (A: 0.1%) 0.27 ± 0.25 (A: 0.5%)
Wu et al. [97]	Retrospective, case–control	117	8.4 (6–12)	>3 years 4.54 ± 1.40 (cases) 4.11 ± 1.21 (controls)	−0.31 ± 0.26 (cases) −0.90 ± 0.30 (controls)	Not measured
Polling et al. [93]	Prospective	77	10.3 (7.1–13.5)	1 year	−0.1 ± 0.7 (A: 0.5%)	25.54 ± 1.35 # (A: 0.5%)
Polling et al. (II) [94]	Prospective, clinical efficacy study	124	9.5 (5–16)	3 years	−0.25D IQR = 0.44 (A: 0.5%)	0.11 mm IQR = 0.18 (A: 0.5%)
Clark et al. [95]	Retrospective, case-control	32	10.2 (6–15)	1 year	−0.1 ± 0.2 (A: 0.01%) −0.6 ± 0.2 (controls)	Not measured
Joachimsen et al. [96]	Observational, cross-sectional case series	56	11 (6–17)	1 year	−0.40 ± 0.49 (A: 0.01%)	Not measured
Zhu et al. [98]	Effectiveness study, prospective, clinic-based	660	9.11 ± 0.09 (A: 1%) 9.19 ± 0.14 (placebo) (6–12)	4 years	−0.41 ± 0.23 # (A: 1%) −0.75 ± 0.64 # (placebo)	0.19 ± 0.13 (A: 1%) 0.40 ± 0.16 (placebo)
Diaz-Llopiz et al. [99]	Randomized, placebo-controlled	200	10.4 ± 2.5 (A: 0.01%) 10.1 ± 2.2 (placebo) (9–12)	5 years	−0.14 ± 0.35 (A: 0.01%) −0.65 ± 0.54 (placebo)	Not measured
LAMP [37]	Randomized, placebo- controlled, double-masked	383	8.4 * (4–12)	2 years	−0.55 ± 0.86 (A: 0.05%) −0.85 ± 0.73 (A: 0.025%) −0.12 ± 0.85 (A: 0.01%)	0.39 ± 0.35 (A: 0.05%) 0.50 ± 0.33 (A: 0.025%) 0.59 ± 0.38 (A: 0.01%)

Abbreviations and symbols: SEP (D): spherical equivalent progression (in diopters), AL (mm): axial length (in millimeters). A: atropine concentration. *: the number was approximated. #: the result was not statistically significant (*p* > 0.05). Notes: in the study by Wu et al., cases were treated with atropine 0.05%, or with 0.1% if progression after 6 months was higher than −0.5 D. In the study by Zhu et al., subjects received atropine 1% once a month for 2 years, then atropine 1% once every 2 months for 1 year, then no treatment for 1 year. The table shows the results of the trials that the authors deem most impactful on future studies and the clinical management of myopia, as well as articles that tried to extend those results to childer of other ethinicities.

**Table 2 pharmaceutics-14-00900-t002:** Percentage of eyes with SEP <0.5 D and >1 D in the ATOM and LAMP trials.

Study Name	% of Eyes with SEP <0.5 D				% of Eyes with SEP >1 D *			
	Placebo	SG 1	SG 2	SG 3	Placebo	SG 1	SG 2	SG 3
ATOM 1 [90]	16.1%	65.7%	-	-	63.9%	13.9%	-	-
ATOM 2 [91]	-	63%	58%	50%	-	18%	18%	18%
LAMP [37]	27.5%	52.7%	32.0%	22.0%	12.5%	9.1%	7.0%	19.2%

Abbreviations: SG = study group. Notes: in this table, study group 1–3 in the ATOM 2 trial refer to atropine 0.5%, 0.1% and 0.01%, respectively. In the LAMP trial, placebo refers to the atropine switchover group, and study group 1–3 refer to atropine 0.05%, 0.025% and 0.01%, respectively. * = in the LAMP trial this column refers to progression >2 D.

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
