# Peer review of "The Role of Atropine in Preventing Myopia Progression: An Update"

_pharmaceutics, 2022, doi:10.3390/pharmaceutics14050900_

Round 1

Reviewer 1 Report

Page 6: “The mean progression before the treatment was -1.05 ± 0.37 D” should be corrected to “The mean progression before the treatment was -1.05 ± 0.37 D per year”

I suggest including (either as columns in Table 1, or as a new table) the results of the studies as percentage of eyes that have progressed 0.5 D and 1 D (as already presented in text).

Page 11: The commercial name of contact lenses is written in 3 different manners: “MySight”, “Misight” or “MiSight”. Please correct.

Page 11: The phrasing “In any case, atropine has shown promise among antimuscarinics” is confusing, perhaps it should be rephrased or even discarded.

Author Response

1.1, 1.3: Since this is a narrative review, we did describe systematically all clinical trials on atropine in preventing the progression of myopia. The authors selected the ATOM and LAMP clinical trials because, in our opinion, these had the most impact on both clinical practice and the design of later studies. We did not include papers where multiple strategies were investigated, e.g. atropine drops combined with orthokeratology, because we wanted to avoid possible confounding factors. As for other articles, some were included for their historical relevance, while others because they tried to extend the results of the trials to children of different ethnicities. However, we would gladly consider including any specific paper if, according to the reviewers, it would make a significant contribution to our review. 

1.2: The paper has been added to our introduction.

1.4: We have now specified the rationale for choosing those studies.

Reviewer 2 Report

The paper provides a discreptive review of a problem valid on everyday basis in clinical ophthalmological practices, which refers to prevention of myopia progression in children. The manuscript provides the possible pathomechanism of development of that condition and different approaches to its treatment.

The manuscript gives a communicative review of on up-to-date studies that tested different dosing of topical atropine for prevention of that condition. I believe that it this description puts in order different concepts of myopia development and modern approach to its prevention. Authors analyze randomized studies and also trial that were not randomized, which is acceptable, as randomized trials on the subject are rather scrace. Authors just need to specify the way they chose these studies for the review.

I have the following specific remarks:

  1. It is a nice descriptive review. However, authors need to precise the rationale of choosing the studies that they included in the review.
  2. Introduction: please note about functional consequences of myopia as well ex. : https://doi.org/10.1155/2019/2654170
  3. Page 3: What is the rationale for presenting these specific 3 trials ? Please expand.
  4. Table 1. Please precise in the table subtitles what was the rationale behind the choice of these studies.

Author Response

point 1: The sentence has been corrected.

point 2: We have included the table as suggested.

point 3: we have corrected the spelling.

point 4: we have rephrased the sentence. It should now be less confusing.

Reviewer 3 Report

The manuscript The role of Atropine in preventing Myopia Progression: an updateby Chierigo et al is a review that discusses the pharmacokinetic, pharmacodynamic, clinical, and tolerability profile revealed first by the multicenter, randomized ATOM 1 and 2 trials and, more recently, by the LAMP Study.

  1. Results Section1. Antimuscarinic Eye Drops – Is there a reason the authors mention only 3 out of the 19 clinical trials that are testing the effectiveness of various concentrations of atropine?
  1. Figure 1. is mentioned in the text but missing from the paper.
  2. Please be consistent with the style of references.
  3. Please proofread for grammatical and syntax errors throughout the manuscript.

Author Response

Point 1: Since this is a narrative review, we did describe systematically all clinical trials on atropine in preventing the progression of myopia. The authors selected the ATOM and LAMP clinical trials because, in our opinion, these had the most impact on both clinical practice and the design of later studies. We did not include papers where multiple strategies were investigated, e.g. atropine drops combined with orthokeratology, because we wanted to avoid possible confounding factors. As for other articles, some were included for their historical relevance, while others because they tried to extend the results of the trials to children of different ethnicities. However, we would gladly consider including any specific paper if, according to the reviewers, it would make a significant contribution to our review. 

Point 2: we removed the mentioning of figure 1.

Point 3 : we checked the paper for inconsistencies in the style of references. We used Mendeley reference manager to automatize the list of references. Please communicate any further specific inconsistencies with our references.

Point 4: we checked for grammatical and syntax errors. Please let us know if there are any other specific grammatical or syntax mistakes.